# Cryo-Electron Microscopy of BfpB Reveals a Type IVb Secretin Multimer Adapted to Accommodate the Exceptionally Wide Bundle-Forming Pilus

**DOI:** 10.3390/pathogens14050471

**Published:** 2025-05-13

**Authors:** Janay I. Little, Pradip Kumar Singh, Montserrat Samsó, Michael S. Donnenberg

**Affiliations:** School of Medicine, Virginia Commonwealth University, Richmond, VA 23298, USA; janay.little15@gmail.com (J.I.L.); pradip.singh@vcuhealth.org (P.K.S.); montserrat.samso@vcuhealth.org (M.S.)

**Keywords:** secretins, type IV pili, cryo-EM, AlphaFold, protein structure

## Abstract

Type IV pili (T4Ps) are multifunctional surface fibers essential for bacterial motility, adhesion, and virulence, found across Gram-negative and Gram-positive bacteria and archaea. Detailed descriptions of T4P structural biology are allowing progress in understanding T4P biogenesis. Secretins, large outer membrane channels, are crucial for T4P extrusion in Gram-negative bacteria. Using cryo-EM and AlphaFold, we modeled the structure of BfpB, the secretin of the Bundle-Forming Pilus (BFP) of enteropathogenic *Escherichia coli*. BfpB exhibits a unique 17-fold symmetry, correlating with the thicker BFP filaments, and diverging from the 12–15 subunits typical of T4P, type 2 secretion (T2S), and type 3 secretion (T3S) systems. Additionally, we identified an extended β-hairpin loop in the N3 domain, resembling features of distantly related T3SS secretins, and an N-terminal helix where a C-terminal S-domain is seen in some T2S and T3S secretins. These findings reveal evolutionary parallels and structural adaptations in secretins, highlighting the link between oligomerization and pilus structure. This work advances our understanding of T4P biogenesis, secretin evolution, and bacterial secretion systems, offering insights into pathogenic diversity and future research directions.

## 1. Introduction

Type IV pili (T4Ps) are the most widespread class of fimbriae, found in both Gram-negative and Gram-positive bacteria as well as archaea [1]. Many pathogens possess T4Ps, including *Clostridioides difficile*, *Escherichia coli*, *Legionella pneumophila*, *Neisseria gonorrhoeae*, *N. meningitidis*, *Pseudomonas aeruginosa*, *Salmonella enterica* serovar Typhi, *Vibrio cholerae*, and *Yersinia pseudotuberculosis*, to name a few [1,2,3,4,5,6,7,8]. They are retractile, long, flexible surface fibers several micrometers in length and approximately 45–100 Å in width [9]. T4Ps are composed of thousands of subunits known as major pilins, and fewer copies of minor pilins that are localized to the pilus tip or scattered throughout the pilus fiber. T4Ps have numerous functions including host colonization, adhesion, biofilm formation, twitching motility, auto-aggregation, and DNA uptake [7,10,11,12]. T4Ps also contribute to virulence in both animal models of infection [7,13,14,15,16,17,18,19] and human challenge studies [20,21]. Assembly and retraction of T4Ps is directed by a complex multicomponent machine that spans the cell envelope including, in Gram-negative bacteria, both the inner membrane (IM) and outer membrane (OM).

The biogenesis of T4Ps requires several key components in addition to the pilin proteins: a prepilin peptidase, a polytopic IM protein, at least one nucleotide-binding protein for pilus extension, a set of proteins that together form a complex spanning the IM, and, in Gram-negative bacteria, an OM secretin through which the pili extend beyond the bacterial envelope [22,23].

Secretins are large multimeric OM channels essential to T4P, type II secretion (T2S), and type III secretion (T3S) systems as well as some filamentous bacteriophages [24]. The T2S system (T2SS), which transports proteins across the OM, shares many structural similarities and homologous proteins with the T4P system, including the secretin [25,26]. The T3S system (T3SS), which delivers effector proteins directly into host cells, shares structural similarities with the T4P system only in that both have secretins [27,28,29]. Similarly, filamentous bacteriophages, such as those infecting *E. coli*, rely on secretin-like proteins, such as pIV, to enable the extrusion of phage particles across the bacterial envelope [30,31]. These secretin homologs form gated channels in the OM, structurally resembling those in T4P systems.

Proteins of the secretin family are typically composed of 12–15 monomers with cyclic symmetry that oligomerize into a ~1 MDa OM pore [27,32,33]. These proteins are highly stable and resistant to heat, detergents and denaturing agents [34,35,36]. Secretin structures reveal plugged central channels, which must open to permit extension of pili or export of other substrates. They have variable N-terminal domains in the periplasm and a conserved C-terminal secretin domain that spans the OM [37]. The secretin domain is rich in β-sheets and forms a β-barrel [33,38,39]. The secretin monomer possesses four β-strands that contribute to the OM β-barrel and gated pore [37,40]. The secretin N-terminal domains are variable in length and sequence. It is speculated that these domains are tuned to the function of their system and the distinct IM and periplasmic components with which they interact. These N-terminal domains include a TonB-dependent transduction domain (N0), multiple heterogeneous nuclear ribonucleoprotein K homology-like (KH-like) domains (N1–N5), and a T4P-secretin-specific amidase N-terminal (AMIN) domain [37,41,42,43]. The N0 domain adopts a βαββαββ fold [32,33,44] but is resolved in very few structures of secretins, as it is thought to be flexible [39]. The N1–N5 domains typically adopt a βαββα fold. Out of these various N-terminal domains, the N0 and N3 domains have been observed in every secretin thus far [45]. Domain organizations of various secretins are summarized in Figure 1A.

While progress has been made in understanding the structure of secretins from T2S and T3S systems, our knowledge of T4P secretins, particularly considering the diversity in pilus structure and function, remains limited. Despite the variable sequence homology among secretin family members (20–70%), these proteins share a conserved architecture, as mentioned above [46]. Recent advancements in cryo-EM techniques have enabled the determination of high-resolution structures for various T4P components, including the secretins [39,47,48,49]. The structural and functional differences between type IVa pilus (T4aP) and type IVb pilus (T4bP) systems, such as variations in pilin size and structure, pilus thickness [1,22,50,51,52,53] and ATPase symmetry [54,55,56,57], highlight the need for a detailed understanding of their unique structural adaptations. T4aP and T4bP shared a common ancestor very early in evolutionary history, but secretins were probably acquired much later [58]. The Bundle-Forming Pilus (BFP) of enteropathogenic *E. coli* (EPEC) is a T4bP that is distinctive for its role in initial adherence to tissue culture cells [59], and the major pilin, bundlin, has a unique fold [60]. Thus, a high-resolution cryo-EM structure of its secretin, BfpB, could reveal unique features that advance our knowledge of T4bP biogenesis and function, particularly in the context of pathogenic bacteria. Therefore, we combined bioinformatics, molecular biology, and cryo-EM to elucidate the structure of BfpB. Our results highlight unique features of this particular T4bP secretin and provide important general insights into the relationship between the number of secretin monomers and the thickness of the pilus.

## 2. Materials and Methods

### 2.1. Bacterial Strains and Plasmids

All strains and plasmids used in this study are presented in Table 1. Bacterial strains were cultured in Luria–Bertani (LB) broth at 37 °C with agitation at 225 rpm. Antibiotics (ampicillin, 200 μg/mL; kanamycin 50 μg/mL) were added to select plasmids or maintain plasmids.

### 2.2. BfpB Protein Purification

For purification of BfpB, *E. coli* strain XL1-Blue (pWS15) was grown at 37 °C in LB broth to an optical density at 600 nm (OD600) of 0.6 and induced with 0.2 µg/mL of anhydrotetracycline at 37 °C for 3 h. Cells were harvested by centrifugation and chemically lysed in buffer A (50 mM tris, 300 mM NaCl, pH 8.0) supplemented with protease inhibitors (Roche, San Francisco, CA, USA), 5% (*w*/*v*) 3-(N, N-Dimethyltetradecylammonio) propane-sulfonate (SB3-14), 10 µg/mL DNase I and 100 µg/mL lysozyme. BfpB was purified by affinity chromatography on Strep-Tactin resin (Qiagen, Germantown, MD, USA) column with buffer A + 0.02% (*w*/*v*) SB3-14. Fractions eluted with 10 mM desthiobiotin were analyzed by SDS-PAGE, combined, and concentrated with 100 kDa (Amicon, Burlington, MA, USA) molecular weight cut-off filter. The concentrated sample was further purified with Superose6 10/300 column (Cytiva, Marlborough, MA, USA) with buffer A + 0.02% (*w*/*v*) SB3-14. Quality of purification and oligomerization were assessed via negative stain TEM.

### 2.3. Cryo-Electron Microscopy

Grids (300 mesh UltraAufoil −1.2/1.3 holey gold with 2 nm thin carbon (Quantifoil, Eberswalde, Brandenburg, Germany) were glow-discharged for 40 s at 25 mA. Purified BfpB, 0.03 mg/mL, 4 μL was applied to the grids. Grids were blotted for 2 s with ash-free Whatman^®^ Grade 540 filter paper in a Vitrobot Mark IV (ThermoFisher Scientific, Waltham, MA, USA) and plunged into liquid ethane. Sample quality and distribution was assessed on a Glacios electron microscope (ThermoFisher Scientific, Waltham, MA, USA). Data acquisition was carried out in a Titan Krios transmission electron microscope (ThermoFisher Scientific, Waltham, MA, USA) operated at 300 kV and in counting mode, with a Gatan K3 detector and a 10-eV slit width Gatan Quantum Energy Filter (GIF). Images were recorded at 54,000× magnification corresponding to a pixel size of 1.68 Å with a defocus range of −1 to −2.25 μM. Datasets were collected in automated mode with the program Latitude (Gatan, Pleasanton, CA, USA) with cumulative electron dose of 50 e^−^/Å^2^ applied over 40 frames.

### 2.4. Single-Particle Image Processing

Movies collected for the BfpB datasets were processed in Cryosparc V4.1. Gain-normalization, movie-frame alignment, dose-weighting, and full and local motion correction were carried out with the patch motion correction function. Global and local contrast transfer function values were estimated from non-dose weighted motion-corrected images using the patch CTF module. Subsequent image processing operations were carried out using dose-weighted, motion-corrected micrographs. One thousand manually picked particles were used to generate templates for automatic particle picking. Further selected particles were processed in various rounds to generate 2D classification and 3D map building. The molecular symmetry was assessed using polar coordinates. Briefly, 3D maps generated from ab initio classification without symmetry (C1) were horizontally sliced, and polar coordinates were measured (i.e., pixel intensities were measured along circular paths at a defined radius) using SPIDER software (version 17.05) [62]. These intensity values were plotted against angular degrees (0–360°) to visualize symmetry patterns.

### 2.5. 3D Model Building and Symmetry Application

An AlphaFold-predicted model of BfpB (PDB: AF-Q9S142-F1_V4, N0 truncated) was used to fit the 3D map and generate a starting model using C17 symmetry. Considering the resolution, we started by fitting the N3 domain where two α-helices were relatively easy to locate into the electron density map. The fit was improved using the Chimera “fit to map” option. For the β-barrel domain, fitting of the β-sheets was guided by the previously published *E. coli* GspD structure (PDB: 5WQ7). Then, C17 symmetry was applied to the fitted BfpB monomer. The gate regions were manually adjusted for optimal fit within the density map. Truncated sections of the BfpB monomer were edited in Phenix to create a single PDB file, which was re-imported into ChimeraX (version 1.8). Individual monomers (A–Q) were saved and combined using ChimeraX’s “combine” command to produce the final PDB file with 17 chains. Further real-space refinement was performed in Phenix, followed by limited manual inspection in Coot to ensure overall model consistency and chain connectivity in regions supported by the map.

### 2.6. Assessment and Purification of Putative Amino-Terminal Domains of BfpB

AlphaFold 2 and other bioinformatic tools such as Swiss Institute of Bioinformatics MyHits motif scan [63] and Rosetta were used to identify the domain regions of BfpB secretins. FastCloning [64] was used to truncate *bfpB* from codons 19–306 (N0 + N3) and 19–202 (N0) in pWS15 and generate pJIL003 and pJIL004, respectively. Thus, we omitted the signal peptide and lipid anchor codons [65] from both constructs. Primer pairs JIL-002 and JIL-003 were used to generate plasmid pJIL004 and primer pairs JIL-002 and JIL-005 to generate plasmid pJIL003. The polymerase chain reaction (PCR) products were digested with DpnI and purified. The purified PCR products and pET28a vector were restriction-digested with NcoI and XhoI nucleases and then ligated. Ligated products were chemically transformed into *E. coli* DH5α competent cells. The truncated *bfpB* fragments were confirmed by sequencing and expressed in *E. coli* BL21(DE3).

For purification of BfpB N0 + N3 and BfpB N0, *E. coli* strains BL21 (DE3) pJIL003 and *E. coli* BL21(DE3) pJIL004, respectively, were grown at 37° C in LB medium to an OD600 of 0.6 and induced with 1 mM of IPTG at 37° C for 2 h. The expression of N0 and N0 + N3 domains was confirmed by Western blot analysis using rabbit polyclonal BfpB antiserum as described earlier [61]. Cells were harvested by centrifugation and lysed by sonication in lysis buffer (50 mM NaH_2_PO_4_, 300 mM NaCl, 10 mM imidazole, pH 8.4) supplemented with protease inhibitors (Roche, South San Francisco, CA, USA). BfpB N0 + N3 and BfpB N0 were purified by affinity chromatography on nickel nitrilotriacetic acid (Ni-NTA) resin (Qiagen, Germantown, MD, USA) column. Fractions eluted with 250 mM imidazole were analyzed by SDS-PAGE, combined, and concentrated with 10K (Amicon, Burlington, MA, USA) molecular weight cut-off filter. Concentrated samples were further purified on a Sephacryl S100 column (Cytiva, Marlborough, MA, USA) with PBS pH 7.4 or 20 mM HEPES, 150 mM NaCl, pH 7.4.

### 2.7. Thermal Stability Assay

The thermal stability of purified BfpB N0 + N3 and BfpB N0 was assessed using the GloMelt Thermal Shift Protein Stability kit (Biotium, Fremont, CA, USA), following the manufacturer’s instructions. A 10× GloMelt dye solution was prepared from the 200× stock and PBS used for size exclusion chromatography, with peak fractions of BfpB N0 and N0 + N3 serving as protein stocks. Reactions were set up in 20 µL triplicates with final protein concentrations of 0.5, 1.0, and 1.5 mg/mL and GloMelt dye. As a positive control, goat β-actin IgG was used at 0.1 mg/mL with GloMelt dye, while the no-protein control contained only GloMelt dye, and the blank was PBS. All solutions were kept on ice in the dark, and reactions were assembled in qPCR plates with optical seals.

Thermal stability measurements were conducted on an Agilent Stratagene Mx3005P qPCR system (Agilent Technologies, Santa Clara, CA, USA), with fluorescence acquisition in the FAM channel (470/510 nm). Starting at 25 °C, the temperature increased by 0.5 °C every 30 s to 95 °C. Fluorescence intensity was recorded at each increment, and readings were corrected by subtracting blank and no-protein controls. The melting temperature was derived from the midpoint of the fluorescence-temperature curve using Microsoft Excel (Microsoft, Redmond, WA, USA).

## 3. Results

### 3.1. Purification of BfpB Multimer

In comparison to a previously reported physical lysis method [66], we found that chemical lysis in the presence of the zwitterionic detergent SB3-14 significantly improved the yield of BfpB-Strep. The initial purification protocol, which employed sonication for cell lysis and SDS for membrane solubilization, yielded 0.2 mg/mL of BfpB from a 2 L culture. Optimization efforts, including the use of a French cell press for homogenization and modifications to the affinity chromatography step, increased the yield to ~0.92 mg/mL. However, chemical lysis with SB3-14 was more productive, resulting in a peak eluate fraction containing ~2.5 mg/mL of BfpB, representing an 11.5-fold increase over the initial protocol. Subsequently, the impact of SB3-14 concentration on BfpB multimer formation and aggregation was investigated. Among various concentrations of SB3-14 (0, 0.007, 0.014, 0.020, and 0.04%), the 0.020% SB3-14 concentration was found to yield the best results, exhibiting the least amount of aggregation and non-oligomerized protein, as assessed by size exclusion chromatography and negative staining (Figure 2). This concentration was therefore selected for subsequent cryo-EM analysis. Nevertheless, the yield of oligomers that were not aggregated remained a limiting factor in data collection.

### 3.2. Cryo-EM Reveals C17 Cyclic Symmetry

The cryo-EM dataset consisted of 9248 movies. Manual and template-based automated particle picking yielded 255,000 particles, out of which 33,096 were usable BfpB particles. Two-dimensional classification yielded various classes, and their 2D averages (a few examples are shown in Figure 3A) showed that most BfpB particles have a toroidal shape with seventeen identical subunits. Their side view reveals density projecting towards the center, and a region of flexibility on one of the toroid’s open sides. The contacts with neighboring particles seen in the raw images did not result in any visible signal around the BfpB averages, suggesting that contacts were random and became canceled upon averaging. Ab initio 3D classification performed without symmetry imposition (C1) resulted in five distinct classes. Of these, three classes exhibited consistent structural features indicative of a uniform architecture with circular symmetry. Collectively, these three classes accounted for approximately 80% of the total particle dataset, with ~56% of particles displaying clear C17 symmetry and ~24% exhibiting ambiguous features suggestive of either C16 or C17 symmetry. The remaining ~20% of particles, corresponding to two classes, were classified as unusable due to poor structural quality. The C17 symmetry for the classes with usable particles was confirmed by two complementary methods. First, polar transformation of a horizontal slice through the 3D reconstructions that showed C17 symmetry clearly demonstrated 17 intensity peaks (Figure 3B). These peaks had uniform separation of 21.35 ± 1.09° (mean ± SD), close to the theoretical value of 21.17° per subunit expected for a C17 symmetry. In contrast, 16 peaks could be distinguished in a similar analysis of a slice through a 3D class with ambiguous symmetry. These peaks had more variable 22.26 ± 2.58° (mean ± SD) separation, and in this case, the inter-peak separation had a bimodal distribution. 11 peaks followed the C17 periodicity, but this periodicity was interrupted by larger separations of 26.09 ± 2.13° in the remaining peaks. This variability strongly suggests that the latter class had C17 symmetry but was slightly tilted, resulting in less defined and more separated peaks as the cutting plane transitions to a different plane of the structure (Appendix A). In a second approach, particles from the ambiguous class (n = 7901) were submitted to refinement either with C16 or C17 symmetry. Refinement with C16 symmetry yielded a 3D structure with poorly defined apical domains and a resolution curve showing a local minimum at around 15 Å, crossing the 0.5 FSC threshold. This was not the case when using C17 symmetry, which yielded a 3D reconstruction with well-defined apical domains and without the pronounced local minimum in the FSC curve (Appendix A). We conclude that BfpB has 17-fold symmetry. A total 25,675 particles from the useful classes were used for the 3D reconstruction with C17 symmetry, which converged to a 4.7 Å resolution 3D map at the 0.143 cutoff (Gold Standard) (Appendix A). The 3D map accession code is EMD-70296.

Additionally, we found that BfpB has a preferential orientation that places the top or bottom of the protein on the grids, displaying fewer side views (Appendix A) and resulting in anisotropic resolution [67]. This limitation likely contributed to a lower resolution than would otherwise have been obtained.

The features of the BfpB cryo-EM density map (Figure 4) compared with other secretin maps (EMD 12874, EMD21154, EMD 21152, EMD21153) [30,49], indicate the presence of two domains, corresponding to the N3 and secretin domains. We speculate that the N0 domain is not visible, likely due to expected flexibility in the region. The 3D map showed that the BfpB multimer has a height of ~113 Å (excluding N0 domain) and maximum width of ~160 Å. The N3 domain comprises the base of the molecule and covers ~25% of the total height, whereas the secretin domain contributes to ~75% of the molecule’s total height. The bulge at the middle of the outer periphery of the secretin β-barrel (S* in Figure 4F) suggests the presence of an S-domain. The S-domain is reported to interact with the LPS layer in T2S GspD and T3S InvG [68,69]. The overall structure resembles a wine cask, with the N3 domain forming the base, which has a narrower width (~124 Å). The structure reaches its maximum width in the middle (~160 Å), while the top of the structure tapers slightly, with an outer width of approximately 148 Å. The molecule has a gate-like arrangement located just above the middle of the height and clearly visible from the top of the molecule (Figure 4A,D). Further analysis of a vertical cross-section of the molecule shows an additional inner layer located just below the gate region and above the N3 domain (Figure 4F), which is uncommon among secretins, previously reported only in two T3SS secretins, InvG (6PEE, *S. enterica*) and MxiD (8AXL, *Shigella flexneri*) [70]. The cross-sectional analysis of the gate region showed the presence of two layers which are expected to form the inner and outer gates (gate 1 and gate 2), respectively (Figure 4E,F), and have been reported in other secretins too [30]. The inner diameter of BfpB is ~96 Å at the base of the N3 domain and becomes narrower towards the gate (~88 Å) and after this bottleneck there is a wider opening of ~114 Å.

### 3.3. A C17 Model of BfpB

We used AlphaFold to build the initial molecular model of the BfpB monomer. The AlphaFold-predicted model (AF-Q9S142-F1-v4) for the full-length BfpB sequence (Figure 1B) achieved a high average pLDDT score of 74.98, indicating high confidence in the prediction. This model delineated three distinct domains: N0 (H74–N199), N3 (E200–E305), and the secretin domain (R306–D546). Subsequently, a preliminary C17 symmetric model was generated by fitting 17 copies of this initial model (lacking the N0 domain) to the 3D cryo-EM map of BfpB (Figure 5) using ChimeraX and Phenix. The monomer was tilted ~30-degrees with respect to the vertical for better fit with the structure. The N0 domain was excluded as there was no corresponding density in the 3D structure, probably due to flexibility of this region.

Structural comparisons revealed that the AlphaFold model shares structural similarities with earlier reported secretin structures from T4Ps, T2SS, and T3SS (Figure 1). Interestingly, AlphaFold uniquely predicted an N-terminal α-helix (N21-S45), a feature that has not been reported in any published secretin structure so far. Density corresponding to this helix region can be observed in the cryo-EM map, lending further support for this unique feature. Interestingly, similar densities are observed in two other secretins, GspD and InvG (Figure 1), but they belong instead to a C-terminal S-domain. Of note, sequence alignment analysis did not reveal that the BfpB N-terminal helix sequence has any significant conservation with the S-domain sequences of GspD and InvG (Appendix A). The S-domains are expected to stabilize the pore complex by interacting with neighboring protomers as well as interacting with pilotin proteins [69]. It is unclear whether the N-terminal alpha helix in BfpB might have a similar role, although the BFP system does not appear to have a pilotin. BfpB is a lipoprotein that is palmitoylated at Cys 18 and thus does not need a pilotin to reach the OM [66]. These findings indirectly support the presence of the N-terminal helix as modeled, as it situates Cys18 within the location predicted for the OM, based on surface hydrophobicity and homology with other secretins (Figure 6A).

The AlphaFold model of BfpB also showed a β-hairpin loop (K214-D253) as an extension of the N3 domain, which makes an additional layer of beta barrel and a curtain-like structure below the gate. Interestingly, this feature has only been reported in two distantly related T3SS secretins (InvG and MxiD). Correspondingly, our cryo-EM 3D map also shows density at the predicted location of this extended N3 domain beta hairpin loop (Figure 4E,F). Although the function of this layer remains unclear, its loop has a serine-rich composition that may create a hydrophilic environment for the pilus as it is extended and retracted.

Although the relatively modest resolution of the final map (4.7 Å) limits precise atomic modeling in such a large and complex assembly, several key features of the AphaFold predicted BfpB model can be reliably fitted and interpreted. These include the novel N-terminal α-helix, the extended N3 domain β-hairpin loop, and the overall cyclic arrangement of the BfpB multimer. Together, these findings not only highlight the structural uniqueness of BfpB among secretins, but also provide a framework for future investigations into its functional roles within the BFP system.

### 3.4. Confirmation of N0, and Stability of the N0-N3 Domain Pair

The above structure analysis using AlphaFold [71,72] as well as other bioinformatics tools such as Swiss Institute of Bioinformatics MyHits motif scan [63] suggest that BfpB has conserved N0, N3 and secretin domains, as shown in Figure 1B. However, while the MyHits motif scanner identified the region 201–306 as “N2 domain”, we believe that this is an N3 domain as N2 and N3 domains share a βαββα fold and all secretins described to date have an N3 domain, but some lack an N2 domain [30,32,33,44,73]. Both prediction tools suggested that the N3 domain ends at Arg306, and the secretin domain (β-barrel) begins at Thr309. Based on this information, and excluding the initial 18 residues that constitute the signal peptide and lipid anchor [65], the putative domains were cloned for expression and purification as follows: N0 (residues S19-T202), N3 (T202-R306), and N0 + N3 (S19-R306). The fragments encoding the N0, N3, and N0 + N3 domains were successfully cloned into plasmid vectors.

Both N0 and N0 + N3 were purified by Ni-NTA affinity chromatography and size exclusion chromatography (Figure 7A,B). Thermostability of these purified proteins was assessed via GloMelt. The melting temperatures of N0 and N0 + N3 were ~50 °C and ~55 °C, respectively (Figure 7C). These results imply that BfpB N0 and N0 + N3 are both folded and stable and they define protein domains. In contrast, we did not observe expression of the N3 domain alone.

## 4. Discussion

Here, we present the cryo-EM structure of the type IVb pili secretin protein BfpB from EPEC. The 4.7 Å resolution map was derived from 25,675 particles extracted from 9248 movies collected on a Krios EM with a K3 detector. The map showed that BfpB is composed of a secretin domain with an N3 domain and that it has C17 cyclic symmetry. The BfpB molecule has a maximum diameter of ~160 Å with two apparent gates, which, upon opening, would be expected to form a pore of ~88 Å in diameter. We fit the BfpB monomer structure predicted by AlphaFold and Rossetta into the cryo-EM map, which, despite the limited resolution, confirms several unique features of the BfpB secretin. The model showed a unique additional layer of β-sheets in the interior of the barrel just below the gate regions and an N-terminal helix over the outer periphery of the secretin domain. Importantly, our cryo-EM map also showed electron density corresponding to these regions (Figure 4E,F). Further, we utilized the predicted model to mark the boundaries for the N0 domain and the N3 domain and measured the domain stability of pure, recombinantly expressed N0 and N0 + N3 domains of BfpB.

The 4.7 Å cryo-EM map provides significant insights into the architecture of BfpB, revealing a barrel-like structure with a unique 17-fold cyclic symmetry (Figure 3). This symmetry indicates that BfpB has the greatest number of monomers reported for any secretin to date, surpassing the 12–15 subunits previously observed in secretins associated with T4P, T2S, and T3S systems (Table 2). We performed multiple rounds of 2D classifications with the same selected particles, modifying the number of total target classes in each attempt to promote re-mixing and segregation. In each case, the prevailing symmetry obtained was C17. Similarly, we used the ab initio 3D model building approach in CryoSPARC with the selected particles, grouping the particles into 3–5 3D classes without applying any symmetry. The results from both 2D and 3D classification revealed C17 as the predominant class, while initially suggesting that some particles might have C16 symmetry. To explore other potential symmetries, we performed 3D reconstruction and refinement using the same set of particles, but with various applied symmetries ranging from C12 to C22, as typically expected in secretin molecules (Appendix A). Although minor improvements were observed in the numerical values of the Gold-standard Fourier shell correlation (GSFSC) resolution (before masking), the overall map quality deteriorated in symmetries other than C17. Analysis of the angular distance between density peaks in classes that showed clear C17 symmetry and those that were initially ambiguous with regard to C16 or C17 symmetry confirmed the C17 symmetry and suggested that apparent ambiguity is the result of a slight tilt of these particles. The high number of monomers that comprise the BfpB secretin is likely an adaptation to meet the specific functional requirements of the EPEC BFP system. Not surprisingly, the C17 symmetry of BfpB results in a multimer with a greater diameter than any secretin reported to date (Table 2 and Figure 4D and Figure 8A). This in turn is consistent with the exceptional thickness of the BFP [60] (~85 Å) even among T4bP systems, which are generally thicker than the filaments of T4aP systems (~45–70 Å) (Table 2). This structural divergence appears to be an essential adaptation to allow extrusion of wider pili, as shown in Figure 8B.

Another distinctive feature in the BfpB cryo-EM map is the additional inner layer, observed below the gate region in the cross-section analysis (Figure 4). Interestingly, the predicted BfpB models showed a unique beta hairpin loop (K214-D253) corresponding to this region. This loop has not been reported in any T4P secretins so far. However, it has been observed in two T3SS secretin proteins, InvG and MxiD from *Salmonella* and *Shigella*, respectively. The existence of similar extended β-hairpins in the N3 domains in certain T3SS secretins is surprising, given the lack of homology between T4P and T3S systems. The evolutionary history of secretins is more complex than that of the most fundamental elements of T4P machinery such as the pilin, extension ATPase, and polytopic inner membrane proteins [58]. Secretins were likely added long after these components, as T4P and T2S systems evolved in Gram-negative bacteria and there was a need for the pilus or substrate to cross the OM. It is likely that the secretin was added independently in different T4P systems and therefore that secretin phylogeny does not parallel that of other components. This may explain why BfpB has features described thus far only in T3SS secretins. Alternatively, this β-hairpin, along with the Ser-rich flexible loop in the additional β-sheet layer, may represent convergent evolution to solve similar functional challenges across different secretion systems. Though there is no function reported for this region, we speculate that it may contribute to structural stability or play a regulatory role in substrate gating.

Despite the relatively low resolution of our BfpB structure, the validity of the findings is strongly supported by several factors. In general, the cryo-EM density map aligns with the AlphaFold-predicted model of BfpB (AF-Q9S142), including the unexpected feature of the additional β-sheet layer made by the extended β-hairpin in the N3 domain and the N-terminal α-helix region (N21–S45). The N-terminal helix is particularly surprising, as it occupies the space where S-domains have been previously reported at the C-termini of the GspD T2SS secretin from *E. coli* and the InvG T3SS secretin of *S. enterica* [68,69]. This N-terminal alpha helix has not been previously reported in any secretin. Both the N-terminal helix and the additional β-sheet layer were confirmed in the model, providing confidence in the structural interpretation. Importantly, insights from the model allowed us to define domain boundaries, which guided the successful expression and purification of soluble and stable N0 and N0 + N3 N-terminal domains. These experimental results reinforce the reliability of the cryo-EM map, despite the resolution. The overall agreement between cryo-EM data and computational predictions highlights the potential of combining these methodologies to decipher challenging protein structures, particularly those with unique or unconventional attributes.

The characterization of BfpB holds substantial importance in the context of bacterial secretion systems. First, it broadens our understanding of the diversity and structural adaptations of secretins. The unprecedented C17 symmetry and unique features of BfpB emphasize the plasticity of secretin architecture in response to functional and evolutionary pressures. These findings highlight how secretins can evolve to accommodate the specific demands of their respective secretion systems, such as handling wider pili in the case of T4bP systems. Second, this work underscores the potential of integrating cryo-EM with computational tools like AlphaFold to tackle challenging protein structures, particularly when limited resolution or protein complexity present obstacles. The success of this approach in the case of BfpB demonstrates its broader applicability to other complex proteins and systems. Finally, understanding the unique features of BfpB provides valuable insights into the mechanisms underlying T4bP function, from pilus assembly to substrate extrusion, which are critical for bacterial pathogenesis. These insights may pave the way for the development of novel therapeutic strategies targeting secretin structures in pathogenic bacteria.

## 5. Conclusions

In conclusion, this study represents a significant step forward in our understanding of secretin architecture and function. The structural characterization of BfpB reveals novel features, including its C17 symmetry, broader pore diameter, and additional inner β-sheet layer, which collectively distinguishes it from other known secretins. These adaptations are likely tailored to the unique requirements of T4bP systems, which generally have wider pili in comparison to other T4P systems. By leveraging cryo-EM and AlphaFold, this study provides a robust framework for future research into the structural and functional diversity of secretins, setting the stage for further investigations into their roles in bacterial secretion systems and potential as therapeutic targets.

## Figures and Tables

**Figure 1 pathogens-14-00471-f001:**
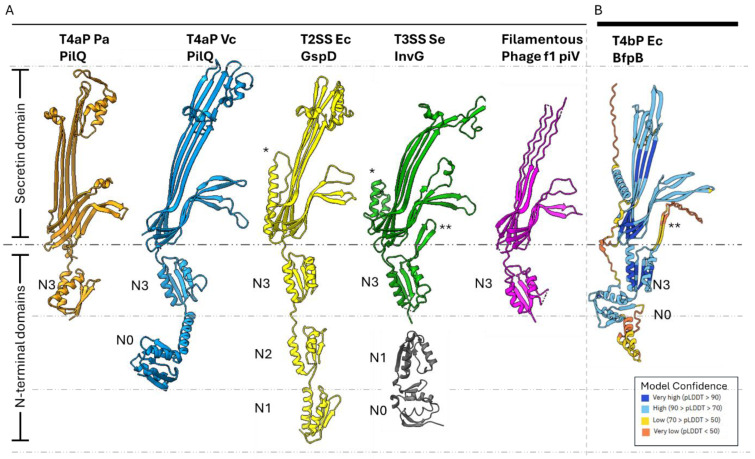
Overview of secretin features and domains of secretin-reliant systems. (**A**) Monomers of secretin proteins from T4P, T2S, T3S, and filamentous phage systems are shown with their domains marked. The secretin domain spans the OM and the periplasmic region, is highly conserved, and consists of 4 β-strands per monomer. The N3 domain precedes the secretin domain and resides in the periplasm. The domain organization is otherwise variable but usually begins with the N0 domain. However, N0 has not been solved by cryo-EM in most structures, except for the T4aP PilQ of *V. cholerae*. Structures in figure: *P. aeruginosa* PilQ (6VE3, orange), *V. cholerae* PilQ (6W6M, blue), *E. coli* GspD (5WQ7, yellow), *S. enterica* InvG (6PEE for N3-secretin domains, green, 6PEM for N0-N1 domains, gray), and f1 phage pIV (7OFH, magenta). (**B**) AlphaFold-predicted structure of EPEC BfpB (PDB: AF-Q9S142-F1-v4). BfpB model colors correspond to pLDDT (model confidence, higher is better) scores for the respective region in the model, as shown. * Indicates S domain. ** Indicates N3 β-hairpin.

**Figure 2 pathogens-14-00471-f002:**
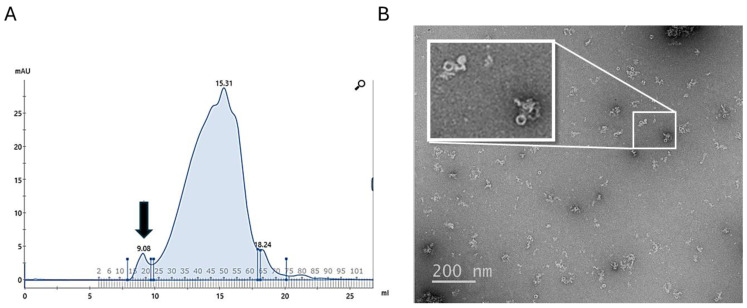
(**A**) Elution profile after size exclusion chromatography with 0.020% of SB3-14 of BfpB previously purified by affinity chromatography. The peak at 9.08 mL marked by an arrow corresponds to the oligomeric form of BfpB, while the peak at 15.31 mL likely represents monomeric or other forms of BfpB. (**B**) shows a micrograph of a negative stained sample from the oligomeric peak, and the inset shows an enlarged view of a selected area showing the toroid’s top and side views. A small degree of random association between neighboring particles is also observed in the electron micrograph.

**Figure 3 pathogens-14-00471-f003:**
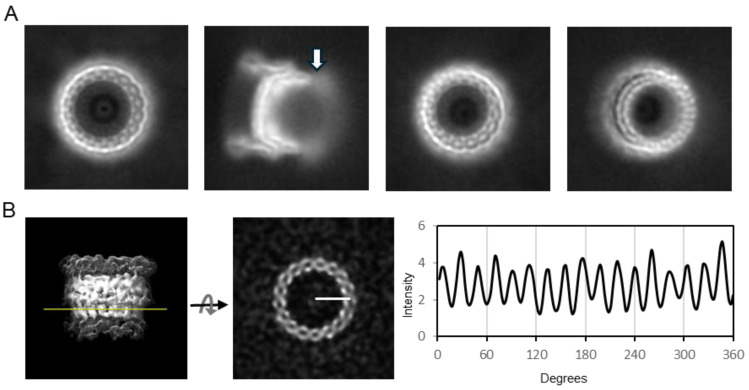
2D averages and symmetry analysis of BfpB. (**A**) Representative 2D class averages showing a top view, a side view, and tilted views. A hazy area showing a region of flexibility in the side view (arrow) likely corresponds to the N0 domain. (**B**) A side view of the ab initio 3D reconstruction (C1; no symmetry applied); the mesh displays the isosurface (Appendix A) at a lower threshold. The yellow line indicates the position of the horizontal slice. The intensity along a circular path at the marked radius in the slice was plotted against angular degrees (0–360°), revealing periodicity consistent with C17 symmetry.

**Figure 4 pathogens-14-00471-f004:**
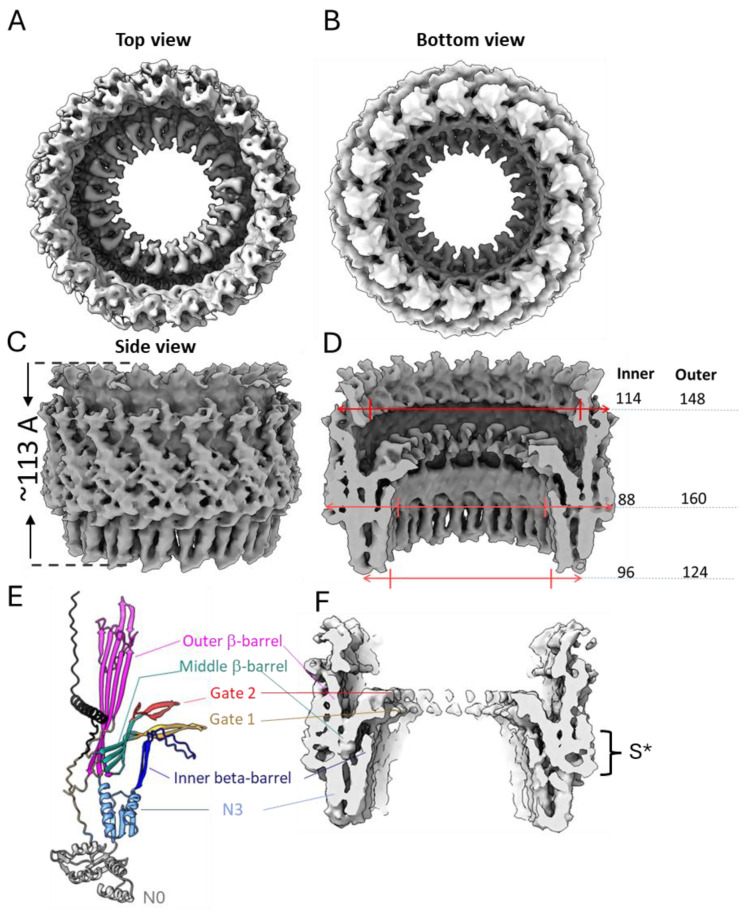
Three-dimensional reconstruction of BfpB (EMD-70296) shown as top (**A**), bottom (**B**) and side (**C**) views along with respective measurements in Angstroms (**D**). (**E**,**F**), side view sliced through the center along with BfpB predicted structure from AlphaFold (PDB: AF-Q9S142-F1-v4) aligned to the side of experimental density. The N-terminal helix, which is expected to occupy the place of the S-domain, is shown in black in model (**E**) and marked with S* on the map (**F**). The N3 domain (201–303) is colored in sky-blue except for the extended N3 β-hairpin loop (215–252), shown in dark blue. The components of the secretin domain (307–545) are colored as follows: outer β-barrel, magenta (307–392, 495–545), inner β-barrel, cyan (393–405, 431–460, 480–492), gate 1, orange (406–430), and gate 2, red (461–479). Note, the N0 domain shown in gray is not resolved into the map.

**Figure 5 pathogens-14-00471-f005:**
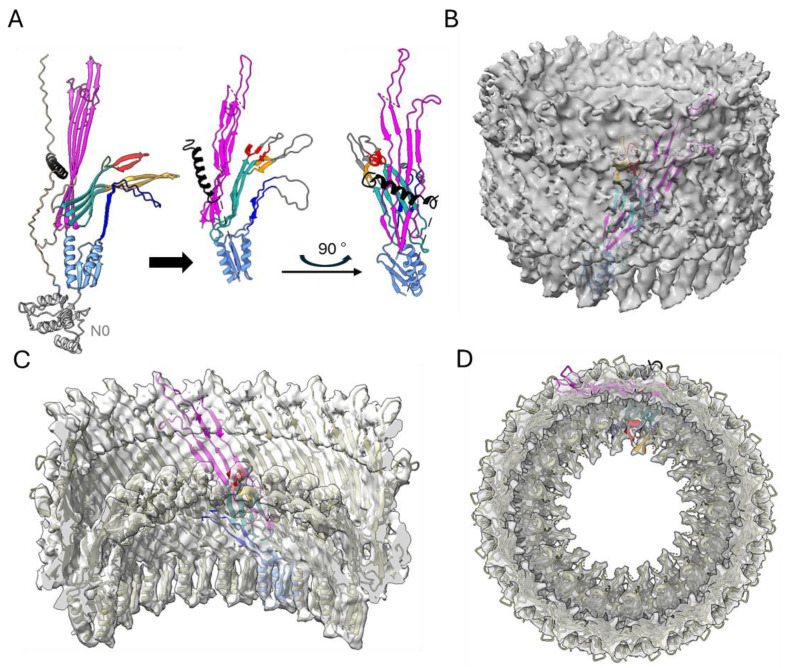
Three-dimensional model building (**A**) BfpB AlphaFold model (Q9S142), colored as described in Figure 4E. The straight arrow indicates modifications made to fit the AF-BfpB model into the map. Briefly, the N0 domain was removed, and the N3 domain, gate 1 (red), gate 2 (orange), and N3 β-hairpin loop regions were rotated to optimally fit into the map. The model on the right offers a 90° turn view of the fitted model. A single unit is shown in color as in Figure 4E; however, density corresponding to β-hairpin loops (226–241, 411–426, and 468–475) present in gates was not seen (likely due to expected flexibility) in the final map, colored in gray. (**B**) The modified model fitted into the cryo-EM map. (**C**,**D**) Inside and top views of the multimer.

**Figure 6 pathogens-14-00471-f006:**
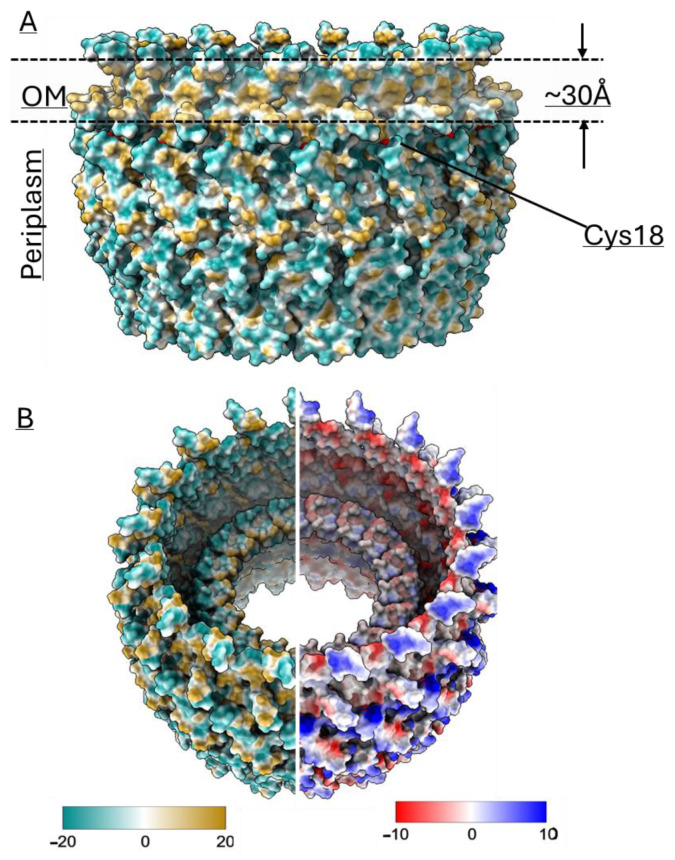
(**A**) A hydrophobicity map of the BfpB model shows an external hydrophobic ring of ~30 Å near the upper border of the molecule, likely corresponding to the location of the OM lipid bilayer. There are small hydrophobic patches also observed in the periplasmic region, which may indicate the area of interaction with the alignment complex proteins of T4P machine. The position of the acylated Cys 18 residue from a single monomer is indicated. (**B**) (**Left**) Hydrophobicity distribution analysis from the inner area of the β-barrel showed the barrel is mostly hydrophilic from inside except for the upper side of gate 2, which is hydrophobic. (**Right**) The electrostatic analysis showed the pore is mostly negatively charged whereas the outer side is positive. Default coloring palette options and ranges for the lipophilicity and coulombic surfaces are used as shown in the respective color scale bars. The lipophilicity display unit is logP (octanol-water partition coefficient) and the coulombic electrostatic potential is displayed in kcal mol^−1^·e^−1^ at 298 K, which is calculated from atomic partial charges and coordinates according to Coulomb’s law. Values were calculated using ChimeraX.

**Figure 7 pathogens-14-00471-f007:**
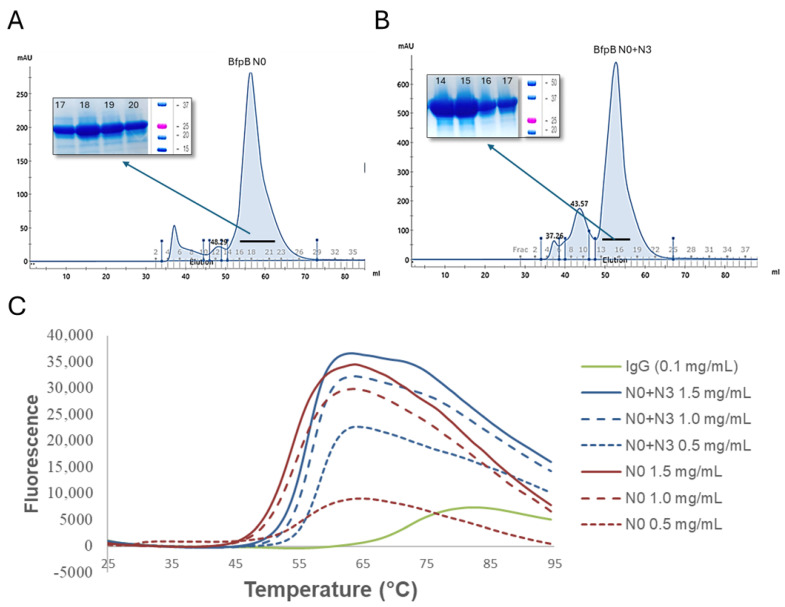
(**A**,**B**) Purification of BfpB N0 and N0 + N3. Ni-NTA affinity chromatography purified samples were further purified by size exclusion chromatography. Corresponding peaks were collected and analyzed on SDS page (inset), BfpB N0 (~21 kDa) and BfpB N0 + N3 (~32 kDa). (**C**) Thermal stability of BfpB N0 and N0 + N3. Control IgG has a T_M_ of 75 °C while BfpB N0 and N0 + N3 have a T_M_ of 50 and 55 °C, respectively. Fluorescence increases with protein unfolding, indicating denaturation, though specific patterns were varied, which we assume depend on protein characteristics.

**Figure 8 pathogens-14-00471-f008:**
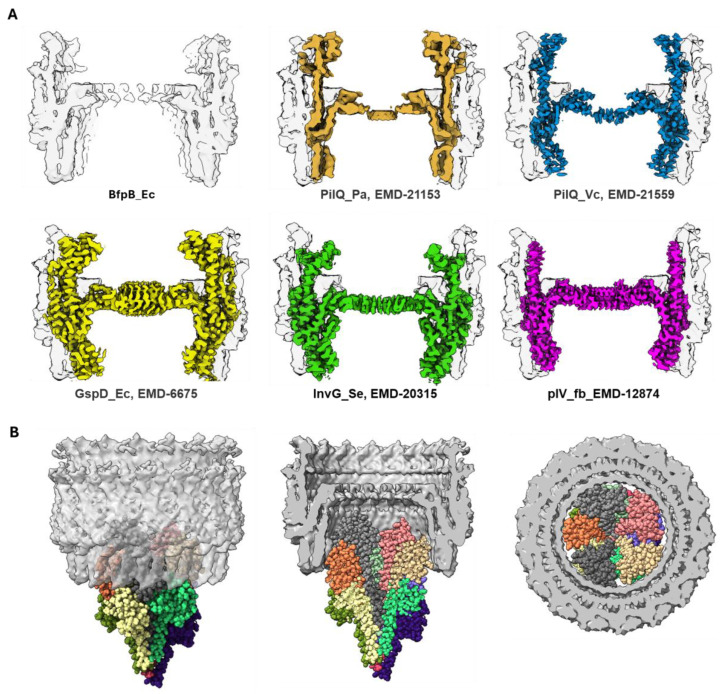
(**A**) Comparison of the cross-section of BfpB (shown in solid gray in the image at top left and as a transparent layer superimposed in the other images) with other reported secretin maps, BfpB (this work; EMD-70296, PilQ_Pa (EMD-21153), PilQ_Vc (EMD-21559), GspD_Ec (EMD-6675), InvG_Se (EMD-20315), and pIV_fb (EMD-12874). (**B**) The BFP shown in space filling model, with different colors for different bundlin monomers) fitted into the inner pore of the BfpB multimer below the gate. The images on the left, middle, and right show BFP through a transparent BfpB secretin from the side, the cutout side view, and horizontal cross-section (below the gate) of BFP fit into the BfpB pore, respectively.

**Table 1 pathogens-14-00471-t001:** Strains, plasmids, and primers used in this study.

Strain and Plasmid	Genotype or Description	Reference
*E. coli* XL1-Blue	*recA1 endA1 gyrA96 thi-1 hsdR17 supE44* *relA1 lac* [F’ *proAB lacIqZΔM15* Tn*10* (TetR)]	Stratagene, La Jolla, CA, USA
*E. coli* BL21 (DE3)	*F-*, *dmc*, *ompT*, *hsdS(rB-mB-)*	Novagen, Burlington, MA, USA
*E. coli* DH5α	*supE44ΔlacU169(φ80dlacZΔM15) hsdR17 recA1 endA1 gyrA96 thi-1 relA1*	Gibco-BRL, Carlsbad, CA, USA
pWS15	*bfpB-Strep* gene cloned into pASK-IBA3	[61]
pJIL003	*bfpB_19-306_* with C-terminal His tag cloned into pET28a NcoI and XhoI sites	This study
pJIL004	*bfpB_19-202_* with C-terminal His tag cloned into pET28a NcoI and XhoI sites	This study
**Primer Name**	**Sequence (5′-3′)**	
JIL-002	GGCACCATGGGATCGGGTAATGGATTTTATAAAGATAATCTTGGCG	
JIL-003	CCGTCTCGAGAGTTTCCTCGTTTGAAAAAGCAATCC	
JIL-005	GGCCTCGAGTCTTTCAAGCTGTGCATTCAGTGTATTAATATATTCG	

**Table 2 pathogens-14-00471-t002:** Summary of secretin and substrate diameters. Diameters of secretins are based on the electron density maps from reported cryo-EM data. The outer diameter corresponds to the outermost points of the secretin β-barrel. The inner diameter corresponds to the smallest diameter below the gate region. Substrate diameter corresponds to the diameter of the following: pilus (T4P), pseudopilus (T2SS), needle (T3SS), and virion (filamentous phage). Reference or PDB code in parentheses. Pa, *P. aeruginosa*; Tt, *T. thermophilus*, Vc, *V. cholerae*, Ec, *E. coli*, Se, *S. enterica*, and Sf, *S. flexneri*.

System Type	Secretin Name	No. of Subunits	Outer Diameter (Å)	Inner Diameter (Å) #	Substrate Diameter (Å)	Sources
T4bP	BfpB_Ec (EMD-70296) *	17 *	~160 *	~88 *	85 (1ZWT)	[60]
	TcpC_Vc	Unknown			80 (3HRV)	[74]
	PilN_Se	Unknown			100 (1Q5F)	[75]
T4aP	PilQ_Pa (EMD-21153)	14	112	~79	60	[49]
	PilQ_Tt (EMD-3985)	13	115	~70	45–70	[33]
	PilQ_Vc (EMD21559)	14	122	~77	60	[39]
T2SS	GspD_Ec (EMD-6675)	15	120	~73	70	[69]
	GspD_Vc (EMD-6676)	15	119	~73	65 (cholera toxin)	[69]
T3SS	InvG_Se (EMD-20315)	15	125	~72	70 (6ZNH)	[68]
	MxiD_Sf (EMD-15701)	15	~138	~78 at most	--	[70]
Filamentous Phage	pIV (EMD-12874)	15	121	~77	60 (2C0W)	[30]

**#** Inner diameter at the narrowest point between N3 domain and gate, which we considered as a bottleneck for their respective substrates. * This work.

## Data Availability

The electron density map is available at the worldwide protein data bank (wwpdb.org) under submission ID: EMD-70296 (accessed on 22 April 2025). The original data presented in the study are openly available in Open Science Framework at https://osf.io/w2587/ (accessed on 30 April 2025).

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
