# Peer review of "Cryo-Electron Microscopy of BfpB Reveals a Type IVb Secretin Multimer Adapted to Accommodate the Exceptionally Wide Bundle-Forming Pilus"

_pathogens, 2025, doi:10.3390/pathogens14050471_

Round 1
Reviewer 1 Report
Comments and Suggestions for Authors
Janay Little et al. presented the T4bP secretin system, which features a unique 17‑fold cyclic symmetry, using cryo‑EM and AlphaFold. This study contributes to understanding the bacterial secretion system and offers insights into a potential therapeutic target. However, I have some reservations about the manuscript, detailed point by point below:
- The authors used carbon layer grids from Quantifoil, yet only 33,096 particles were obtained from 9,248 micrographs. This low yield may result from damage to the carbon film during glow‑discharging. I recommend that the grid quality be carefully assessed before sample application.
- While the pixel size and defocus values are provided in Table S1, it would be preferable to include this key information in the Methods section as well.
- In line 236, the manuscript should report 255,000 particles, as indicated in the Table, rather than the number after selection. Furthermore, the particle counts for different classes and the final number used to construct the final map should be provided. Notably, Table S1 lists the particle count for 3D refinement as identical to that for 2D selection, which is unlikely given the existence of different classes (e.g., the C16 class) and the reported 45–55% for the C17 class. I recommend that the authors double-check these numbers.
- In Figure 6, the units of the color scale bars should be clearly indicated.
- Considering the high-order symmetry, the authors might consider applying symmetry expansion to improve the resolution. In particular, further analysis of the C16 class—comprising approximately 30% of the particles—could be beneficial.
Reviewer 2 Report
Comments and Suggestions for Authors
This manuscript presents new structural insights into secretin by combining cryo-electron microscopy (cryo-EM) with AlphaFold model predictions. The authors demonstrate that BfpB assembles into a unique C17-symmetrical ring characterized by a broader pore diameter and an additional inner β-sheet layer. This work serves as an excellent example of how integrating experimental data with in silico predictions can elucidate macromolecular structures.
Detailed Comments:
- Resolution Map: The authors report a 7.27 Å resolution map of BfpB; however, the corresponding FSC curve is not included. Please provide the FSC data to support this claim.
- Elution Profile (Figure 2a): The manuscript states that oligomeric BfpB elutes at approximately 9 mL on a Superose 6 10/300 column. An elution volume of 9 mL typically corresponds to a molecular size greater than 850 kDa (consistent with a 17-mer BfpB). Please clarify this discrepancy. Additionally, a description of the other peaks is needed, as they appear to dominate the sample.
- Additional Signal (Figure 2b): In Figure 2b, extra signal is observed in association with the circled BfpB. Please provide more information on whether this signal represents the formation of complexes or is due to random protein aggregation, as this clarification would help readers better understand the conformation of BfpB.
Reviewer 3 Report
Comments and Suggestions for Authors
In this manuscript, Little et al., reported cryo-EM structure of BfpB Type IV secrection multimer. The authors used size exclution chromatography, cryo-EM and thermal stability assay to characterize the BfpB multimer. I believe this manuscript is of high quality and is suitable for publication after the addressing the following concerns:
(1) The finding that BfpB forms a 17-mer require more evidence. The authors used maunal inspection of 2D and 3D classes, as well as imposing a C17 symmetry during refinement to show that the BfpB adopts a 17-mer configuration. Because of the inherent low number of particles, as well as strong orientation preference, such analysis may be easily affected by the random sampling during cryo-EM data processing. I recommend authors explore other possibilities of the multimer stoichometry and systematically test the possibility of other multimer states by trying to impose C16, C17 and perhaps other symmetry operators ( see https://www.biorxiv.org/content/10.1101/2024.05.09.593390v1) for more evidence.
(2) As supplementary material, please include the FSC curve, orientation distribution plot information of the 3D map reconstruction.
